# Environmental inefficiencies for arrival flights at European airports

**Xavier Olive[1], Junzi Sun[2]\*, Luis Basora[1], Enrico Spinielli[3]**

**1** ONERA DTIS, Université de Toulouse, Toulouse, France, **2** Faculty of Aerospace Engineering, Delft University of Technology, Delft, the Netherlands, **3** Eurocontrol, Performance Review Unit, Brussels, Belgium

\* j.sun-1@tudelft.nl

## Abstract

In this paper, we analyze two months of trajectory data for aircraft landing in five major European airports. Based on open ADS-B data from the OpenSky Network and open performance models, we enrich all trajectories with automatically detected procedure information, fuel consumption, and emissions for supported aircraft types. To assess the inefficiencies associated with holding patterns, point merges, and continuous descent operations across different airports, we propose methodologies to quantify and compare these environmental inefficiencies. Holding patterns are found to have a higher negative impact on the environment than point merge and continuous descent operations. Furthermore, the paper provides recommendations for procedure evaluations of future airports, which could help policymakers and relevant stakeholders to evaluate the environmental performances of arrival procedures based on open data and open models.

## 1 Introduction

Civil air traffic around airports, located for the most part around largely populated areas, is currently subject to noise and local air quality analyses that assess the detrimental impact of aviation on local populations. Aircraft burn fuel and emit pollutants all along their landing and take-off (LTO) cycle, while engines run at various regimes including taxiing, take-off, and climb, as well as approach and landing.

All these steps are subject to a careful analysis of environmental impact. Sustainable aviation objectives call for improvements in all aspects of air traffic operations. Recently, more airports and airlines have been implementing electric push-back [1] or one-engine taxi [2, 3]. Climb and descent flights, especially continuous climb and descent operations, are also carefully monitored and optimized to save fuel and limit emissions [4, 5]. However, there has been limited research comparing emissions at different airports under different arrival procedures. Traffic in Terminal Maneuvering Areas (TMA) includes a lot of legacy procedures, subject to many local geographical, legal, and specific constraints. A wide variety of rules and procedures exist, which take into account runway configurations [6], neighboring civil or military airfields, and air spaces with their associated traffic.

The shift to performance-based navigation allows new procedures to emerge, for example, the point-merge procedures [7] in TMA aim at improving operational performance and

**Data Availability Statement:** Research data are shared publically at https://doi.org/10.4121/20411868.

**Funding:** The author(s) received no specific funding for this work.

**Competing interests:** The authors have declared that no competing interests exist.

decreasing the environmental impact of aviation. Recently, access to large amounts of open access data such as ADS-B data from The OpenSky Network [8], the development of preprocessing libraries [9] and open performance models [10] have offered new analyses and environmental performance assessments for such new procedures.

Several studies have assessed the environmental impact of aviation both at the European level [11] and at the global scale [12] using a descriptive approach, and [13] addresses the environmental impact from an optimization perspective. Other TMA studies, like [14], focus on flight time as the criterion to assess vertical efficiency in descent procedures. At the TMA level, [15] addresses the environmental impacts due to congestion, and [16] discusses various operational and fuel inefficiencies for different airports. Point-merge has been designed with operational performance in mind. However, there have been few studies aiming at the evaluation of emission performances. Furthermore, there is a huge gap in the literature on the comparison of environmental inefficiencies among different airports and procedures, which is exactly the main issue addressed by this paper.

In this paper, we focus on the analysis and quantifying of the environmental inefficiencies in TMA for aircraft arriving at airports. This work is built upon a set of open source data from the OpenSky Network. It uses the *traffic* library [9] to detect arrival procedures like holding patterns and point merges. The environmental analysis is based on the *OpenAP* model [10], which allows the estimation of fuel consumption and emissions based on flight data. In addition, we propose a new method to compare inefficiencies and apply it to a set of standard arrival procedures at five major European airports. Based on the analysis with real data, we also recommend how future simulations and optimizations could be implemented to better design arrival procedures at airports.

The paper is structured as follows. Section 2 presents the data we use to support the analysis. Section 3 details various operational concepts implemented to sequence aircraft in terminal maneuvering areas. Section 4 introduces the methodology used to assess time-based and emission-based inefficiencies, before presenting results in Section 5. In Section 6, we discuss why these procedures incurring in such inefficiencies are still implemented, and how optimal operations can be made a reality. Finally, Section 7 presents the conclusion and further recommendations.

## 2 Dataset description

In this paper, we aim at assessing the environmental inefficiencies of various airports implementing different sequencing strategies for their arrivals. While every airport is unique and has its specificity, we chose the following use cases for this paper, encompassing two months (October and November) of data in 2019. We look at the global environment impact across all modes of operations (nominal, busy, disrupted, etc.) occurring during the scope of the dataset, and focus on global averaged estimations.

Automatic Dependent Surveillance–Broadcast (ADS-B) is a cooperative surveillance technology that provides situational awareness in the air traffic management system. Aircraft determine their position via satellite, inertial, and radio navigation and periodically emit it (roughly one sample per second) with other relevant parameters to ground stations and other equipped aircraft. Signals are broadcast at 1090 MHz; a decent ADS-B receiver antenna can receive messages from cruising aircraft located up to 400 km far away, while the range is much lower for aircraft flying at low altitudes or on the ground.

The data used for this study is collected by the OpenSky Network [8], a network of ADS-B receivers, which offers querying capabilities on their database for academics. Recorded data contains timestamps (added on the receiver side, with many receivers equipped with a GPS

nanosecond precision clock), transponder unique 24-bit identifiers (`icao24`), space-filled 8-character callsigns, latitude, longitude, barometric altitude, geometric altitude, ground speeds, true track angle, and vertical speed. The data has been made available on the 4TU. ResearchData repository [17].

**London Heathrow** `EGLL` is a very busy airport near London, well-known for its holding patterns (Fig 3) stacked nearby the city. Attempts to reduce holding time have led to the definition of extended AMAN (arrival manager), which proceeds by reducing cruising speeds during the final en-route phase of flight, several hundred (usually 180-200) nautical miles away from the airport.

**London City** `EGLC` is a smaller airport near the city's financial district, with specific procedures implemented to fit in surrounding traffic, including a particularly steep gliding profile. London City is one of the airports implementing a point-merge pattern (Section 3.3) above the North Sea.

**Dublin** `EIDW` has been an early enthusiast user of the point-merge pattern over the Irish Sea, since 2012. They currently implement another variation of the pattern when they operate on their West concept, with two point-merge flows being further merged before the final approach (Fig 4).

**Paris Charles de Gaulle** `LFPG` is one of the major airports around Paris, with a dedicated ATC center (Athis-Mons) focused on the sequencing of arrival flows in Paris (hence a particularly big radius for the area of interest). Traffic is mostly vectored, but a point merge is also documented on the North-Western side, mostly for the incoming flow from transatlantic flights.

**Amsterdam Schiphol** `EHAM` is one of the busiest airports in Europe where traffic is mostly vectored to two of the six operating runways. Holding patterns are few and a fair share of trajectories landing at Amsterdam Schiphol operate with continuous descent operations.

## 3 Operational concepts and identifications

### 3.1 Vectoring and tromboning

The most basic way to sequence trajectories before landing is to introduce delay and lengthen the total distance path that the aircraft must fly to ensure an appropriate and safe aircraft separation and to optimize the runway throughput.

Vectoring operations consist mostly in instructing aircraft to change their heading for some time before heading back to documented procedure points. Procedure points may also be bypassed during low activity to optimize flying time. Tromboning is a particular way of vectoring trajectories. Trombones are made of parallel segments, upwind and downwind, with a set of navigational points supporting path stretching, which helps to systematize the traffic flows to the runways. Trombones may be designed and documented as part of the STAR procedures (like in Vienna airport) or implemented at the discretion of ATC as a set of common practices (like in Frankfurt or Zurich airport, see Fig 1).

For this analysis, we considered vectoring and tromboning as unspecific actions to take any trajectory away from its shortest path to the runway. We considered such actions as the baseline and did not make any particular effort to characterize them apart from the total duration of their course through the TMA.

### 3.2 Holding patterns

A holding pattern is a maneuver where an aircraft flies a racetrack-shaped pattern in a designated area. Such a maneuver can be implemented en route, when the crew needs to run through checklists [18] and troubleshoot problems, or by refueling aircraft [19]. They are often

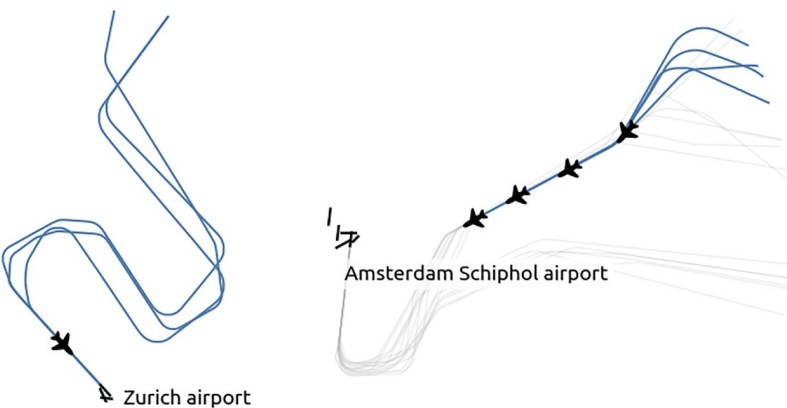

**Fig 1. Tromboning (left) and vectoring (right) are two common techniques to sequence aircraft in terminal maneuvering areas.**

implemented in TMA as a last resort to sequence aircraft using limited space. When operations are disrupted, it is a common practice to stack holding patterns with aircraft flying the race-track shape at various altitudes, the lower aircraft having the higher priority.

Holding patterns are defined from a navigational point, called *holding fix* which forms the end of an inbound leg. Depending on the initial bearing of the trajectory, aircraft enter a holding with different patterns (Fig 2). Holding patterns are mostly flown in a standard direction (right-hand turns) but non-standard patterns are also common (Fig 2c).

Historically, the racetrack shape has been preferred over circles as the latter limit situational awareness. The introduction of RNAV made it easier to fly any pattern, but since the rules of aviation were standardized before GPS came into common use, racetrack patterns developed for holding at the time have remained the norm.

London Heathrow airport has been selected in the dataset because of the many holding patterns implemented during peak hours (Fig 3).

Despite being carefully designed, holding patterns are very hard to properly label systematically due to large variances in the way to enter a holding pattern and in the duration of the straight legs, if any. Attempts to detect circles, or intervals where the track angle covers a range of 360 degrees fail in many corner cases. We relied here on the method implemented in the *traffic* library [9] which is based on a neural network model. The statistics of go-arounds and holdings in the dataset is shown in Table 1.

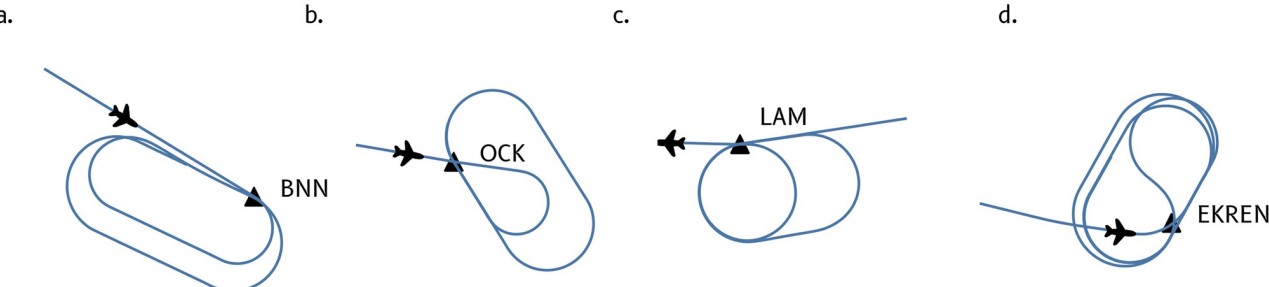

**Fig 2.** Holding patterns may be entered according to different patterns: direct entry (a.), tear-drop entry (b.), and some variants may also be implemented, with some oval shapes becoming circles (c.) or switching from a left-hand turn to a right-hand turn upon entry (d.).

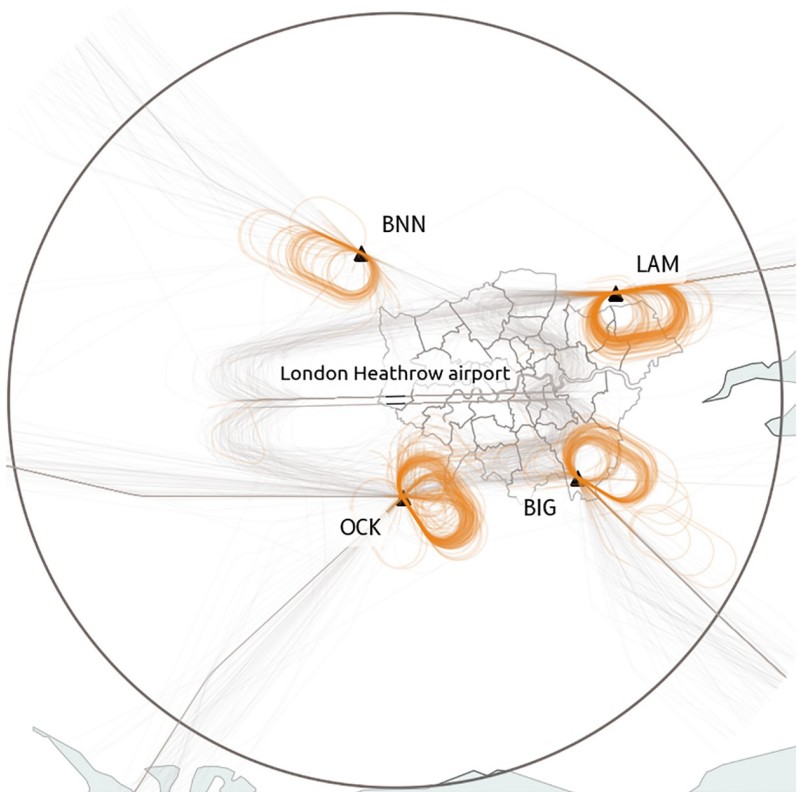

**Fig 3. London Heathrow is well-known for its historical holding patterns (Bovingdon VOR, Lambourne VOR, Ockham VOR, and Biggin VOR) located very close to densely populated London boroughs.** (Public domain map data from Natural Earth: https://www.naturalearthdata.com/).

### 3.3 Point merge

Point merge is a technique for sequencing arrival flows developed by the EUROCONTROL Experimental Centre in 2006 [20]. Point merge relies on a specific route structure, made of a point (the merge point) and arcs equidistant to that point (the sequencing legs). Aircraft fly along the sequencing legs to create spacing, before heading to the merge point with a `DIRECT`, instructing the pilot to head directly to the beacon at the center of the merge point, when spacing is obtained.

Point merge was first deployed in Oslo airport (2011) and Dublin airport (2012) before being adopted by other airports around the world. We focused in this study on a few airports with good coverage on the OpenSky Network and implementing a point merge, which are Dublin (`EIDW`), London City (`EGLC`) and Paris-Charles de Gaulle (`LFPG`), which are shown in Fig 4.

**Table 1. Statistics about datasets used in this study.**

| airport | number of flights | go-arounds | holding patterns |
|---|---|---|---|
| EGLC | 4364 | 37 | 50 (1.2%) |
| EGLL | 38550 | 145 | 13680 (36%) |
| EHAM | 34762 | 65 | 649 (1.8%) |
| EIDW | 17457 | 55 | 4438 (4.5%) |
| LFPG | 37085 | 135 | 78 (2.1%) |

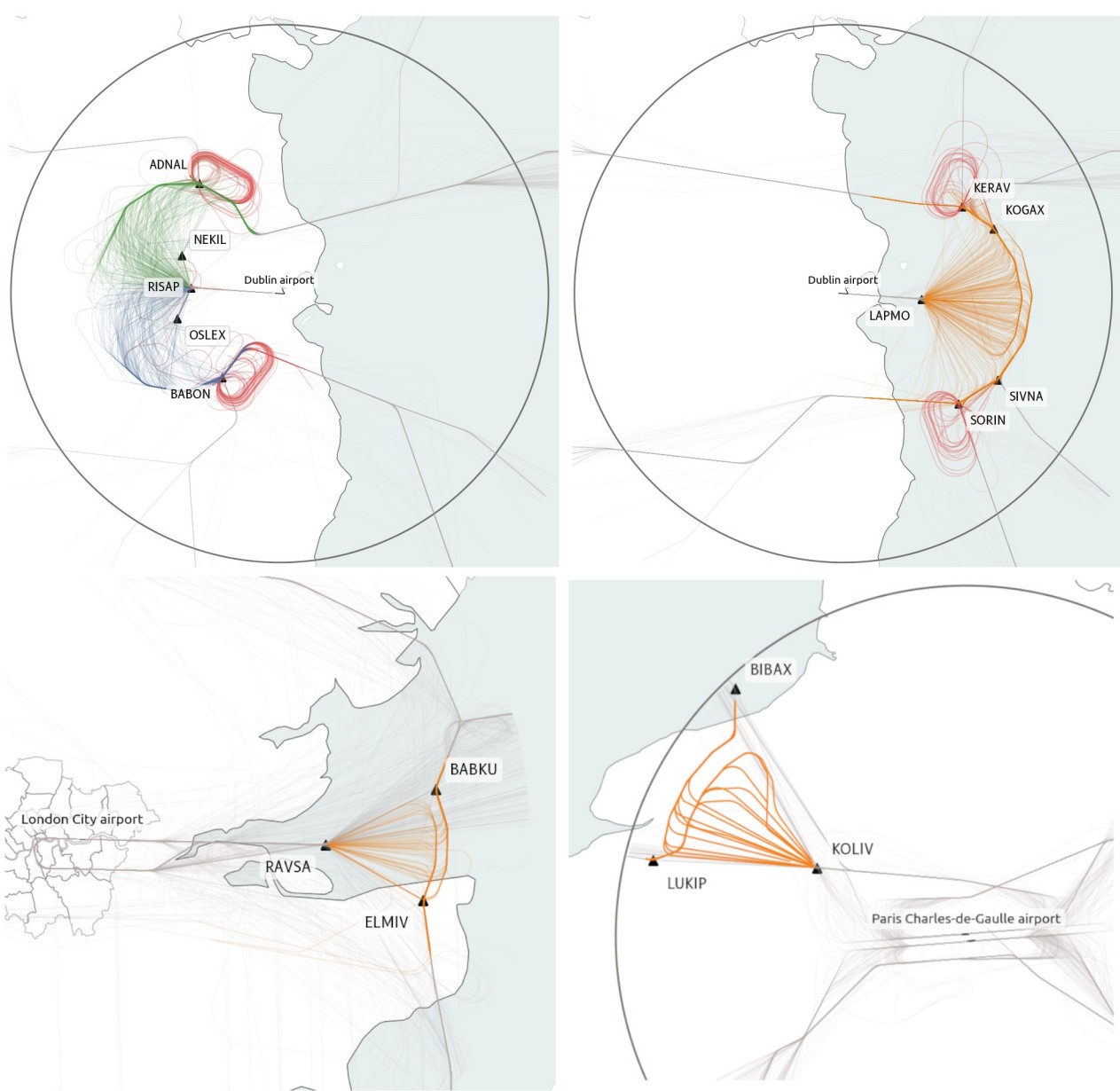

**Fig 4. Point-merge sequencing in Dublin airport.** Top left: runway `10R`, right: runway `28L`, bottom left: London-City airport, bottom right: Paris Charles de Gaulle airport only implements a point merge for its north-west flow. (Public domain map data from Natural Earth: https://www.naturalearthdata.com/).

Point merge patterns tend to emerge during peak hours, but some airports implement them more systematically than others (Table 2). Similarly, locations of merge points depend on the requirements of airports. Dublin and London City use the structure before the final approach, with smaller distances than in Paris, where a single point merge system is sometimes operated to sequence traffic on one single incoming traffic flow (from the North-West, i.e. more or less transatlantic flights).

Various techniques have been used in different studies in the literature [21, 22] to identify point merge structures of real traffic data. We relied on the method implemented in the *traffic* library [9] to identify trajectories flying a well-defined point merge structure. The method is

**Table 2. Point merge implementations across three airports (Dublin, London City, and Paris).**

| airport | runway | merge point | distance (in nm) | number of flights |
|---------|--------|-------------|------------------|-------------------|
| EIDW | 28L | LAPMO | 15/17 | 4438 (25.4%) |
| EIDW | 10R | NEKIL, RISAP | 11 | 378 (2.1%) |
| EIDW | 10R | OSLEX, RISAP | 11 | 275 (1.6%) |
| EGLC | all | RAVSA | 15/16.5 | 545 (12.5%) |
| LFPG | all | KOLIV | 35/40 | 15 (0.04%) |

based on the identification of constant distance legs (with respect to the merge point) followed by DIRECT order to the merge point. DIRECT segments are labeled when the track angle matches the bearing to the target waypoint.

## 3.4 Continuous descent operations

Continuous descent operations (CDO) reduce the vertical inefficiency of approaching flights by eliminating level flight segments. Even though such operations are adopted by main airports to reduce fuel consumption and noise impact, not all CDOs are optimal CDOs, which only occur when idle thrust is applied during the entire descent. Such an operation often requires perfect conflict-free descent trajectories, a precise calculation of the top of the descent, and perfect synchronization with ATC, which is hardly the case in real operations.

Without access to the mass of aircraft, it is hard to conclude whether a descent flight is an optimal continuous descent. However, even the most simple form of CDO, where no level segment is present during the descent, can already bring a positive impact on the fuel consumption [23] and emissions. In this paper, we identify continuous descent flights for all airports by first labeling flight phases contained in each trajectory based on the fuzzy logic proposed in [24]. Then, any trajectory containing level flight segments less than 0.5% of the total duration is considered a continuous descent. Fig 5 shows the examples of a non-CDO trajectory and a CDO trajectory, as well as their identification process.

Table 3 shows the total number of flights in the dataset and the number of continuous descents as a comparison. In particular, Amsterdam and Dublin airports show a large ratio of continuous descents.

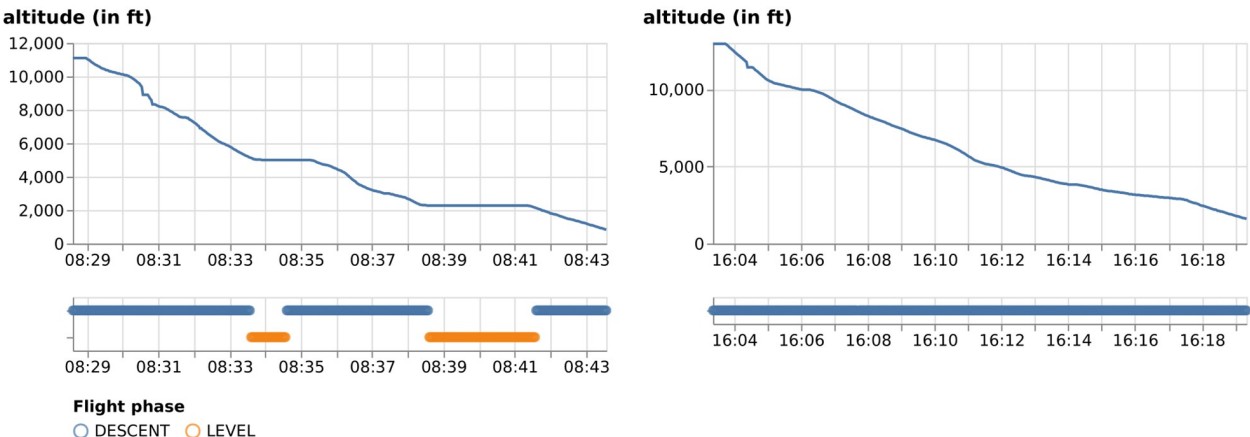

**Fig 5. Identification of non-CDO and CDO flights arriving at EHAM.** The CDO flight is identified as the one without level segments during the descent.

**Table 3. Continuous descent implementation statistics across all airports.**

| airport | number of flights | continuous descents |
|---|---|---|
| EGLC | 4364 | 109 (2.5%) |
| EGLL | 38550 | 5927 (15.4%) |
| EHAM | 34762 | 10820 (31.1%) |
| EIDW | 17701 | 8521 (48.1%) |
| LFPG | 37085 | 3527 (9.5%) |

## 4 Methodology

### 4.1 Large scale processing of data

The dataset presented in Section 2 contains only ADS-B data, i.e. positional information about aircraft with their first derivatives. We enriched the data with several levels of metadata information so as to compare the environmental impact of various approach procedures across different airports.

Origin and destination information is provided by the OpenSky Network Impala database; airports are inferred from the first and last positions of trajectories in the database. Aircraft types are provided by the OpenSky database.

Then we processed every single trajectory with the traffic library in order to:

- label pieces of trajectory with information about point merge, holding patterns, and go-arounds;

- estimate fuel flow and emissions, including $CO_2$, $H_2O$, $SO_X$, $NO_X$, HC, and CO, using OpenAP (see Section 4.2);

- aggregate total information by flight with type codes, runway, distance, duration, holding pattern duration, point merge duration, go around start time, total burnt fuel, and total emissions.

Fig 6 plots a first overview of various metrics for the flights in the dataset. On the left plot, flight duration distributions must be considered together with the size of the area of interest, 50 nm by default, 60 nm for EGLC, 90 nm for LFPG. These sizes are determined based on the radius beyond which the capacity management procedures occur, e.g., point-merges and holding patterns. These are different from each airport and have been selected based on exploration of the flight data and airport arrival procedures. London Heathrow is therefore the most affected airport by operational inefficiencies and delays, with up to one hour spent by aircraft waiting in a 50 nm radius around the airport, mostly with holding patterns. However, we did not take varying levels of traffic into account in this analysis. Considering that London Heathrow, Amsterdam and Paris see a similar traffic load, we consider that averaging over the two months of operation is enough to come to our conclusions.

For a more fair comparison between airports, we normalized all physical quantities by the radius of the area of interest: normalized distances cannot go below one unless data is missing (but we cleared those trajectories from the dataset). A normalized distance of 4 means that a trajectory has been extended so as to fly 4 times the distance of the straight line before landing.

Normalized fuel consumption is proportional to emissions of several pollutants, such as $CO_2$ and $NO_X$. The scores in this plot suggest that London Heathrow performs badly on these metrics. On the other hand, HC emissions (with a similar profile as CO) for London City airports seem high when compared to fuel consumption.

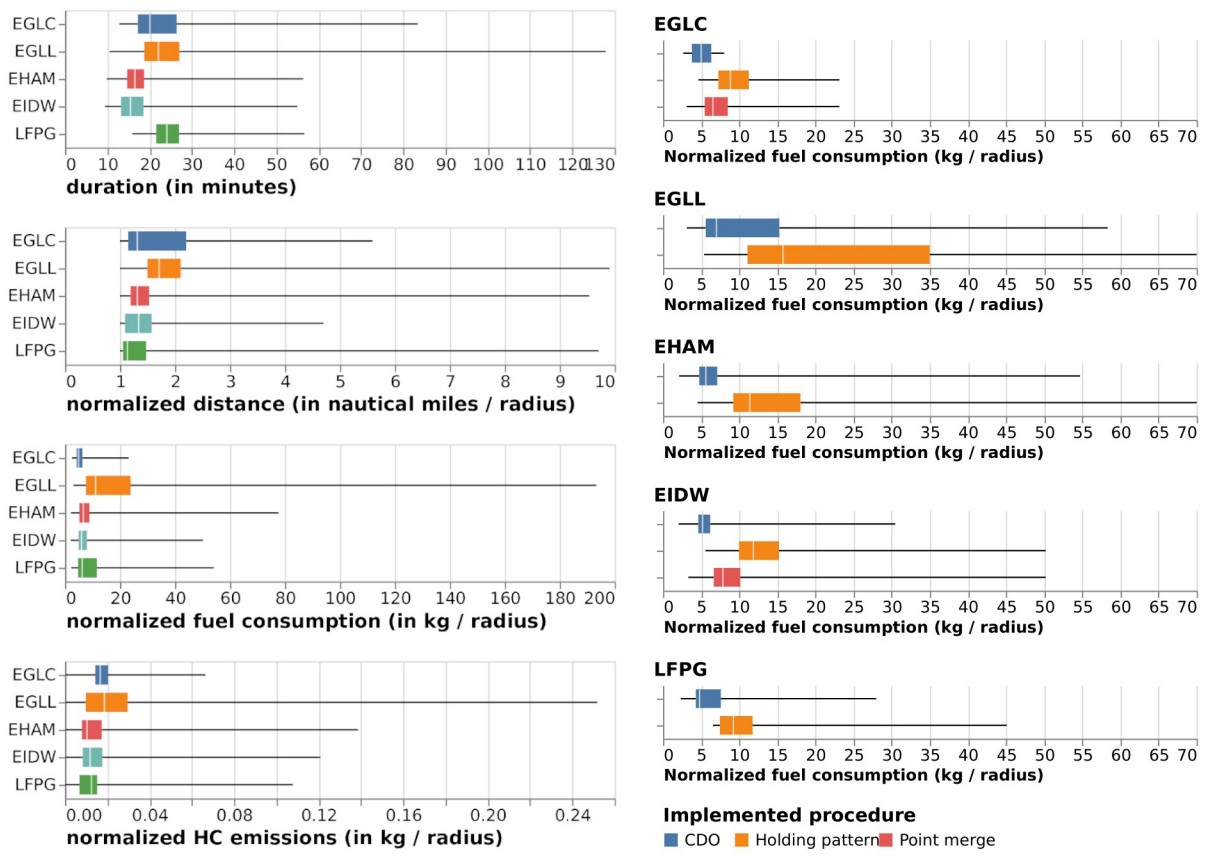

**Fig 6. Overview of the distributions of flight durations, fuel consumption, and emissions over all airports of the dataset (left) and by implemented procedure (right).**

On the right part of the plot, the distributions suggest that among the procedures used to optimize operations in the TMA, holding patterns seem to induce a very high fuel consumption and that continuous descent together with point merge operations seem to require not as much fuel.

## 4.2 Estimation of fuel and emissions

Emissions from all flights are calculated using the OpenAP library, which provides the necessary aircraft performance and emissions models to estimate fuel consumption and emissions based on ADS-B data. Different emission types for common aircraft types are considered, which are $CO_2$, $H_2O$, $NO_X$, $SO_X$, CO, and HC.

OpenAP defines the performance and emission models for more than 20 of the most common aircraft types. For less common aircraft types, we choose a similar type based on a synonym database defined in OpenAP. The calculation of fuel and emissions are all based on the mass that is 90% of the maximum landing weight for the specific aircraft type.

Among all emission types, $CO_2$, $H_2O$, and $SO_X$ are linearly related to fuel consumption. For other types of emissions, OpenAP extends upon the ICAO Aircraft Engine Emissions Databank [25] and Boeing Fuel Flow Method [26]. The ICAO model provides engine emissions from sea-level static tests, while the Boeing model provides emission corrections at different altitudes.

## 4.3 Inefficiency assessments

In the previous section, we showed that capacity and safety procedures, including holding patterns and point merge, increase the flight time during the approach. Consequently, they also cause extra fuel consumption and emissions in the TMA of an airport. In this section, we explain the different approaches to assessing inefficiencies.

The evaluation of environmental inefficiency consists of two parts: fuel inefficiency and emissions inefficiency. Commonly, the emissions include $CO_2$, $H_2O$, $SO_X$, $NO_X$, HC, and CO. The first three of the six types of emissions are linearly related to fuel consumption, whereas the last three emissions have a non-linear relationship with fuel consumption.

To enable the inefficiency calculation, we need to establish a baseline for each capacity measurement procedure. We start with the same airport and separate flights into two different datasets, with or without a certain procedure. In Fig 7, we show the total flight distance and fuel consumption for flights with and without point merge procedures. It is apparent that longer flight distances yield larger fuel consumption and, in turn, lead to higher emissions. The few outliers with low fuel consumption for a very large distance are trajectories with few bogus positional information undetected by the preprocessing: the anomaly in the curviline distance is detected by the fuel consumption calculation which is only based on altitude and speeds.

As a demonstration purpose for the inefficiency assessment, our objective is to study the difference in fuel consumption between flights with and without point merge independent of other factors, such as flight distance or other types of air traffic control procedures. Hence, we need to explore the difference in the following two distributions of fuel consumption, as shown in Fig 8.

We use a bootstrapping approach to draw a large number of random samples (with replacement) from the two distributions and compare fuel consumption differences between these random samples. These differences are then analyzed. For example, Fig 9 illustrates the distribution of fuel inefficiency due to point merge procedures. By running the t-test on the original

**Fig 7. Sampled datasets containing 2000 A320 flights arriving at EIDW, with and without point merge procedures.**

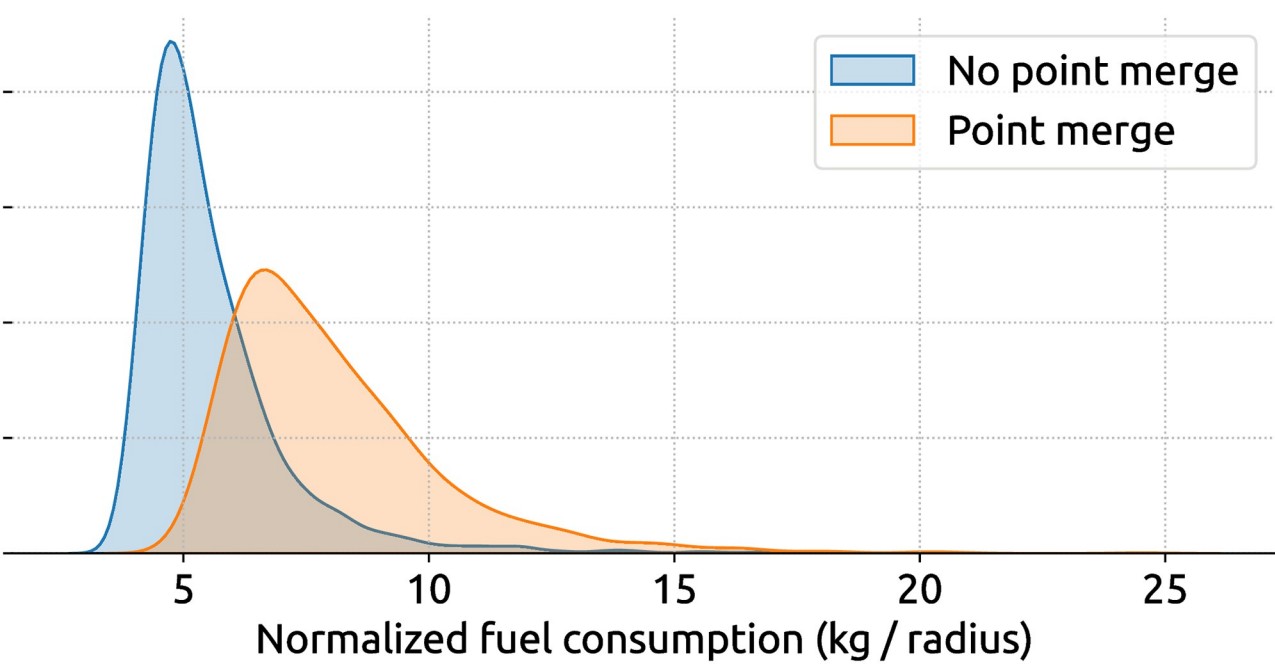

**Fig 8. Distribution of fuel consumption between all A320 flights with and without point merge procedure.**

two distributions, we obtain a t-statistic of 42.8 with a p-value $\ll 0.01$, which shows a significant difference between the two distributions.

Based on Fig 9, by randomly comparing descending flights containing point merge procedures with the ones containing no such procedure, we can conclude that approximately more than 75% of point merge yields a higher fuel consumption. The mean and median normalized fuel caused by the point merge is 3.1 kg and 2.3 kg, resulting in unnormalized fuel consumption of 155 kg and 115 kg, respectively.

The same approach can be applied to other types of comparisons for other procedures like CDO and holding patterns. We can also employ the comparison of fuel and emissions between different airports, as well as different procedures across different airports.

It is worth noting that, in order to cope with the different areas of interest shown in Table 4, we normalized the fuel emission by the radius of the area from each airport. This way, a direct comparison can be made between airports with different sizes of terminal maneuvering areas.

## 5 Results

### 5.1 Comparing procedures

In Fig 9, we already demonstrated the process of comparing the fuel inefficiency caused by point merge. The same analysis can be applied to other procedures. Fig 10, shows the fuel inefficiency caused by holding patterns in EGLL. The mean and median normalized excess fuel is 10.8 kg and 6.2 kg, which is equivalent to 540 kg and 310 kg of excess fuel.

Comparing Figs 9 and 10, we can see between the double and triple amount of excess fuel caused by holding over point merge across two different airports, based on the mean and median excess fuel consumption. To further compare two different procedures, we applied the same inefficiency analysis from holding and point merge from EIDW and showed the results in Fig 11. Here, we see the mean and median of normalized excess fuel consumption per

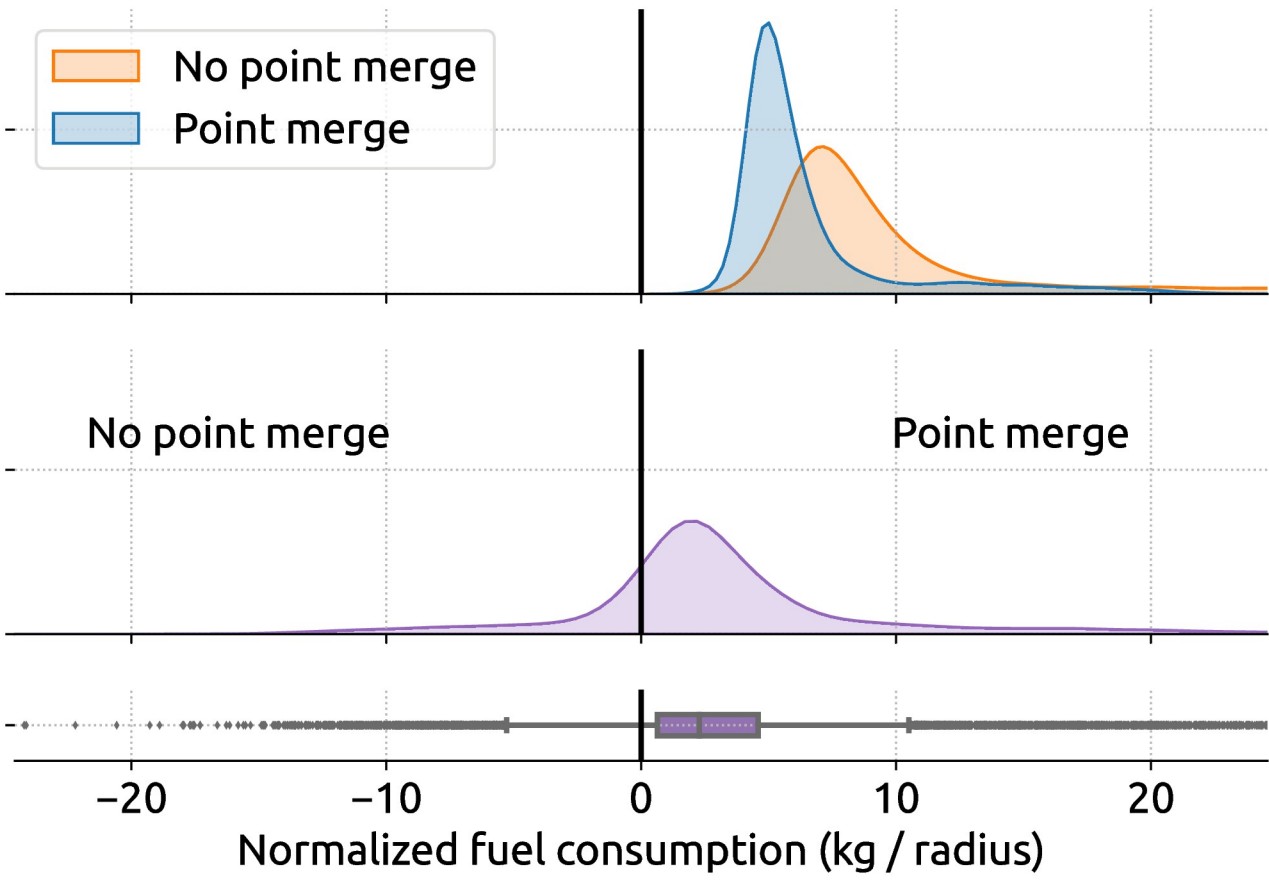

**Fig 9. Distribution (in purple) of fuel inefficiency caused by point merge procedure.** The inefficiency in the second plot is calculated by bootstrapping using the distributions with and without point merges.

holding are 4.2 kg and 3.4 kg, which is equivalent to 210 kg and 170 kg of unnormalized fuel consumption, respectively.

While procedures like holding patterns and point merge have negative effects on fuel and emissions, CDO is designed to minimize excess fuel by optimizing aircraft vertical speed during the descent. Thus, we can quantify the reduction of fuel when compared to traditional non-CDO trajectories. Fig 12 shows the quantity of reduced fuel consumption due to CDO for flights arriving at EHAM. CDO trajectories achieve the mean and median reductions of normalized fuel consumption by 1.6 kg and 1.3 kg. They are equivalent to 80 kg and 65 kg of fuel before normalization.

**Table 4. Description of the datasets used in the study.**

| airport | code | area of interest (radius) | size of the dataset |
|---|---|---|---|
| London Heathrow | EGLL | 50 nm | 38,550 trajectories |
| London City | EGLC | 60 nm | 4,364 trajectories |
| Dublin | EIDW | 50 nm | 17,457 trajectories |
| Paris Charles de Gaulle | LFPG | 90 nm | 37,085 trajectories |
| Amsterdam Schiphol | EHAM | 50 nm | 34,762 trajectories |

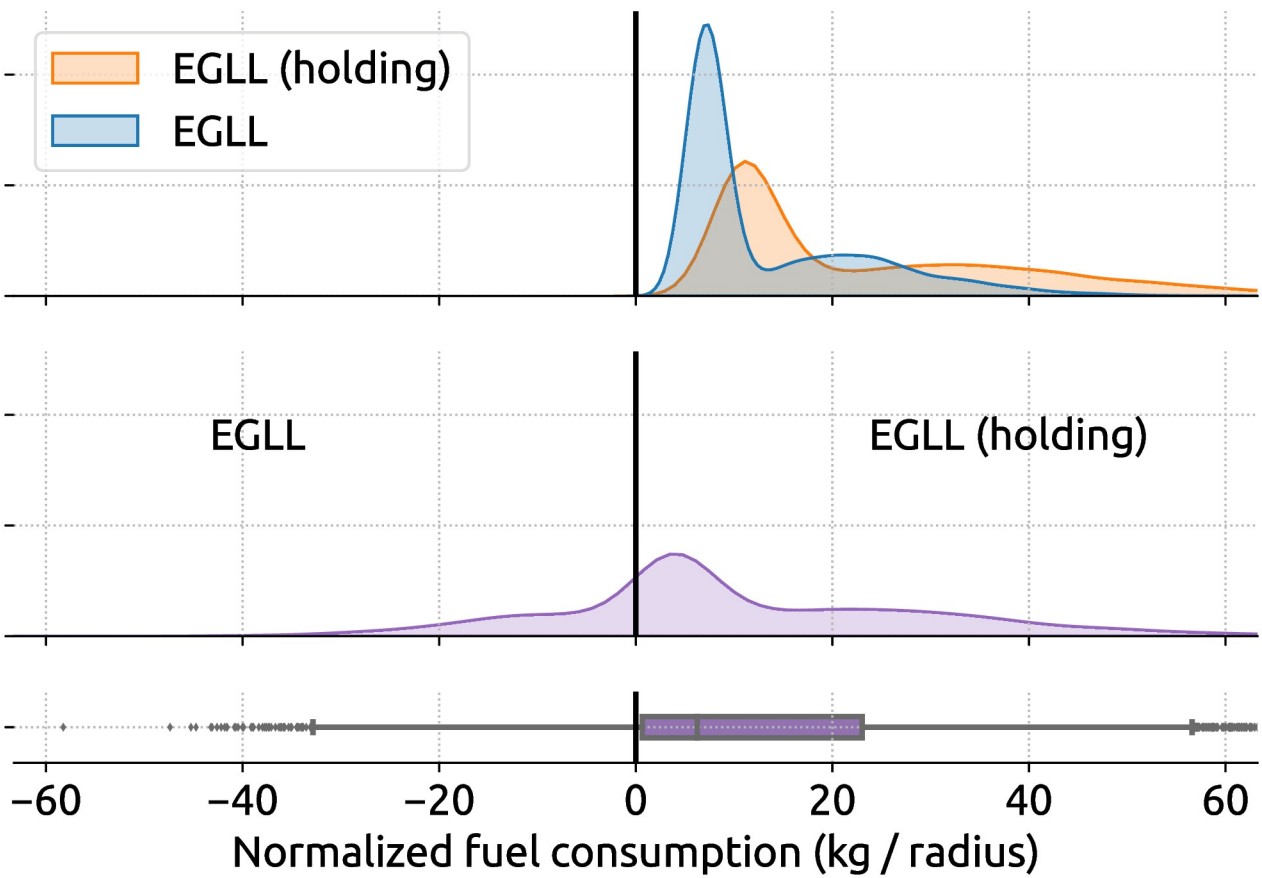

**Fig 10. Fuel inefficiency caused by holdings at** EGLL. The radius for normalization is 50 nautical miles. The mean and median normalized excess fuel consumption per holding is 10.8 kg and 6,2 kg.

### 5.2 Comparing airports

Once estimated fuel and emissions are normalized for the radius of each selected airport, the inefficiencies are comparable among airports. In Fig 13, we show the comparison between EHAM and EGLL, in terms of fuel consumption. We can conclude that EHAM is more efficient. The mean and median normalized fuel savings are 8.5 kg and 4.1 kg, which are around 425 kg and 205 kg when unnormalized.

The comparison is performed for all airport pairs, and the results are shown in Fig 14. In this figure, we illustrate the difference between both fuel and CO emissions. The distributions for $CO_2$, $H_2O$, and $NO_X$ are similar to fuel flow since they can be considered linearly related to fuel consumption during the descent. On the other hand, HC is similar to CO in terms of its non-linear relationships with fuel consumption. Overall, we can observe that EGLL performs considerably worse than all other airports. For the remaining four airports, EGLC performs slightly better in fuel consumption, but not in terms of CO (and HC) emissions. For the rest, all airports other than EGLL perform similarly.

In Tables 5 and 6, we show all the pairwise comparison of excess emissions between airports and procedures. All values are expressed in kilograms for the same distance of 50 nm for all airports.

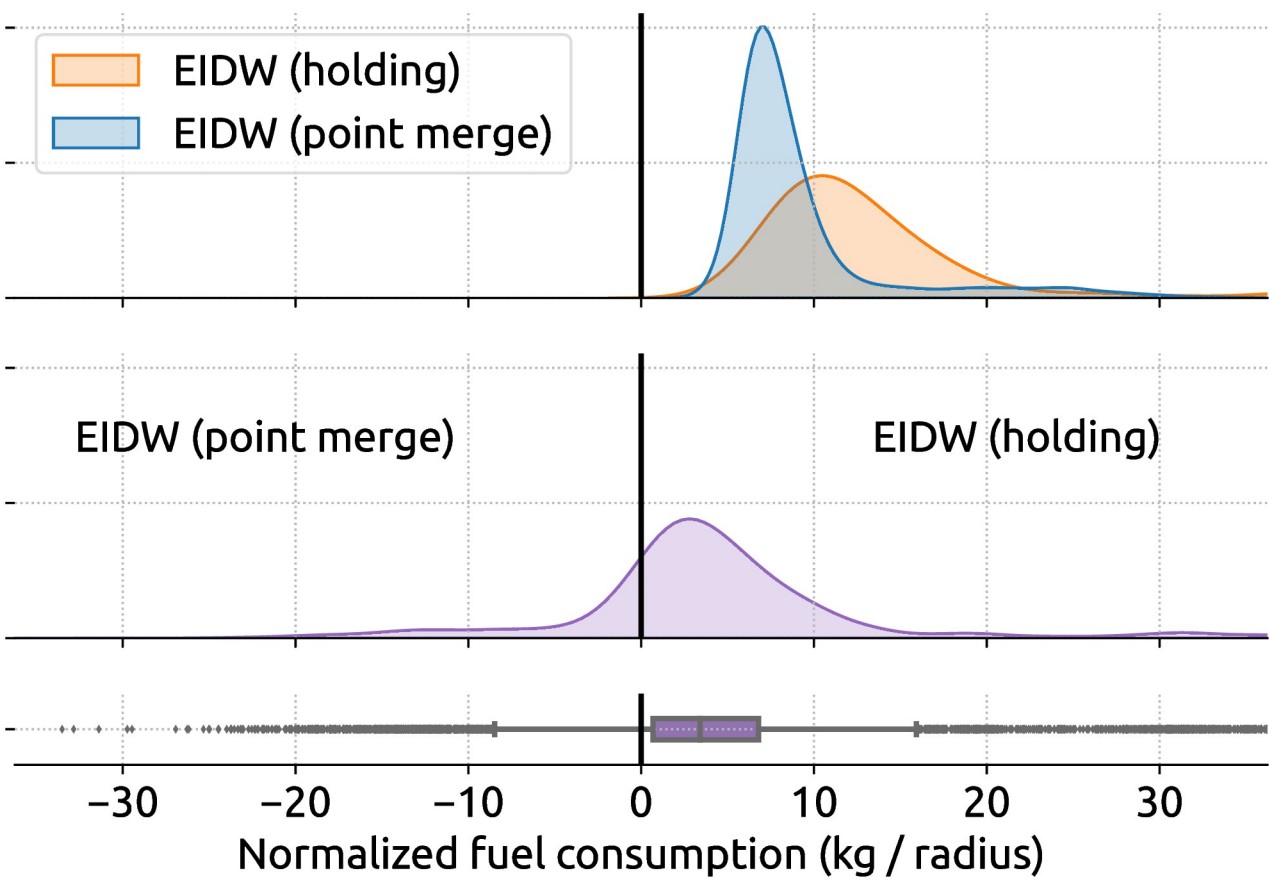

**Fig 11. Comparing the fuel consumption of holdings and point merges at** EIDW. The radius for normalization is 50 nautical miles. The mean and median of normalized excess fuel consumption per holding are 4.2 kg and 3.4 kg.

## 6 Discussion

In this paper, we detailed a systematic approach to assess the environmental impact of inefficiencies for aircraft trajectories arriving at different airports and suggested a bootstrapping approach to compare distributions of fuel consumption and pollutant emissions.

One of the most difficult aspects in the assessment of such inefficiencies is related to the reason why the inefficiencies are introduced in the first place. This type of analysis is applied to real data, which is the result of optimization and sequencing actions from air traffic control. Reasons for implementing such actions are not present in the data, where only the results of the actions are observable.

Inefficiencies are usually the consequence of a safety requirement: procedures causing such inefficiencies are needed to ensure a safe sequencing of aircraft by a decentralized ATC, whose primary objective is the safety of the airspace they manage, while the optimization of flight time is only a secondary concern. Airborne time is usually the most tangible metric for ATC to grasp and actions to optimize it, with a direct impact on runway throughput, are desirable.

However, we showed in previous sections that all strategies to sequence aircraft do not have the same environmental impact in terms of fuel consumption (directly related with $CO_2$, $NO_X$, $SO_X$, $H_2O$) and other pollutants (such as HC or CO). The bootstrapping approach helped compare the impact of such strategies while taking into account all the variability and uncertainty associated with it. Simple vectoring actions were too general to enable a clear analysis

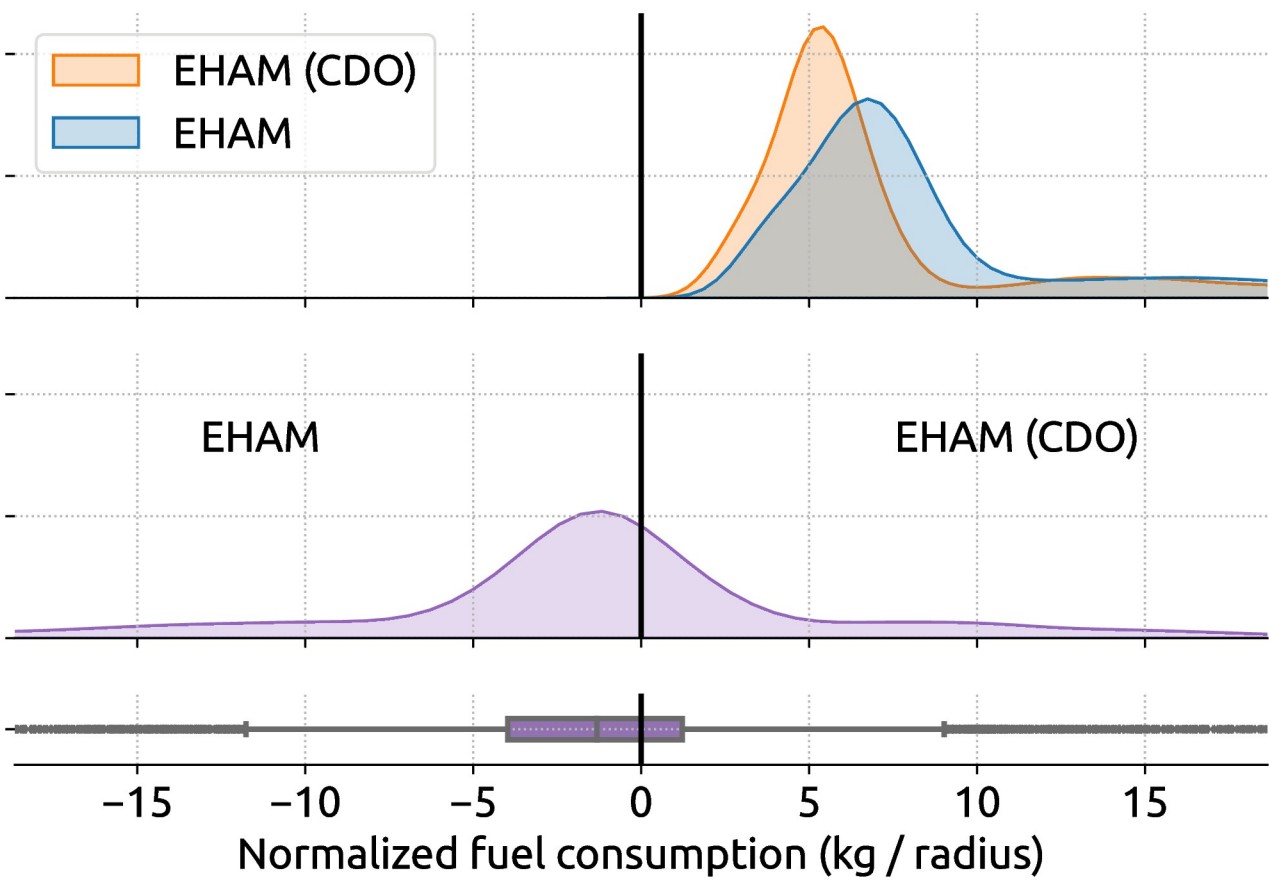

**Fig 12. Comparing the fuel consumption of CDO and non-CDO at** EHAM. The radius for normalization is 50 nautical miles. The mean and median of normalized fuel savings for CDOs are -1.6 kg and -1.3 kg.

but more high-level strategies such as point merge, continuous descents, and holding patterns enabled a clear comparison between procedures.

We analyzed both horizontal (including holding and point merge) and vertical procedures (CDO). Overall, point merges have shown a better emission performance than holdings. In 2016, SESAR JU has adopted the solution of continuous descent operations (CDO) using point merge [27]. However, we can see from the real flight data that CDOs and point merges are rarely combined. For example, in EIDW with most point merges, only 377 out of 4192 point merges (7676 CDOs) are approaches with both CDO with point merge. Judging from the data, we conclude there is still room for improvements in combining both vertical and lateral optimal procedures.

Throughout the paper, the assumption of 90% of maximum landing weight was adopted to enable the same baseline for emission calculations for all flights acrossd different airports. In the absence of real aircraft weight, such simplification to reduce the uncertainty across different airlines. However, it likely introduced bias in the results. In the further research, more accurate estimation of mass at top of descent could increase the accuracy of the analysis.

On the other hand, comparisons between airports should be taken with extra care as they are subject to different operational constraints which do not appear directly in the data. For instance, London TMA airspace is subject to many operational constraints which makes the

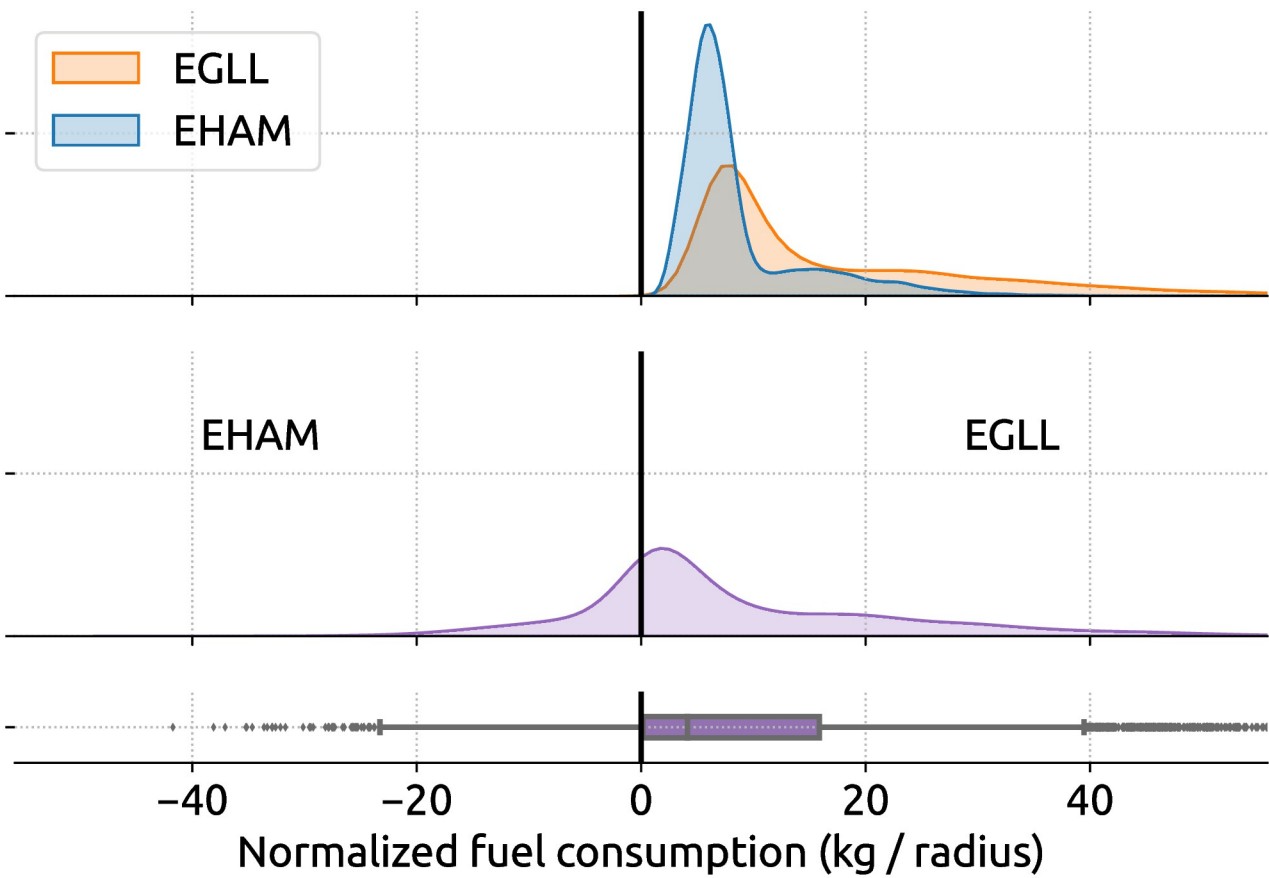

**Fig 13. Comparing the fuel consumption of flights between** EHAM **and** EGLL**.** The radius is 50 nautical miles for both airports. The mean and median of normalized fuel consumption differences are 8.5 kg and 4.1 kg.

use of CDO not practical: the analysis performed in this paper should incentivize airports and ATC to work together on the design of airspaces and descent patterns taking into account these constraints. The simulation-based analysis could provide a better method to compare airport environmental performance and support policies for future operational improvements. Different alternative operational scenarios can be simulated based on initial data provided from existing flights. This way, the environmental inefficiencies of existing strategies and other candidate strategies can be compared based on the methodologies proposed in this paper. The same methodology opens the opportunity to analyze environmental inefficiencies beyond the procedures included in this paper, including, for example, 4D trajectory-based operations and decentralized air traffic control.

## 7 Conclusion

In this paper, we analyzed two months of open trajectory data landing at five different European airports. We estimated fuel consumption and pollutant emissions to compare the environmental inefficiencies of different arrival procedures. We explained why a fair comparison between airports and procedures can be difficult without proper simulations. Despite that, we could still draw some conclusions based on our data analysis. First, holding patterns do have a negative impact on the environment compared to other sequencing strategies. Secondly, we

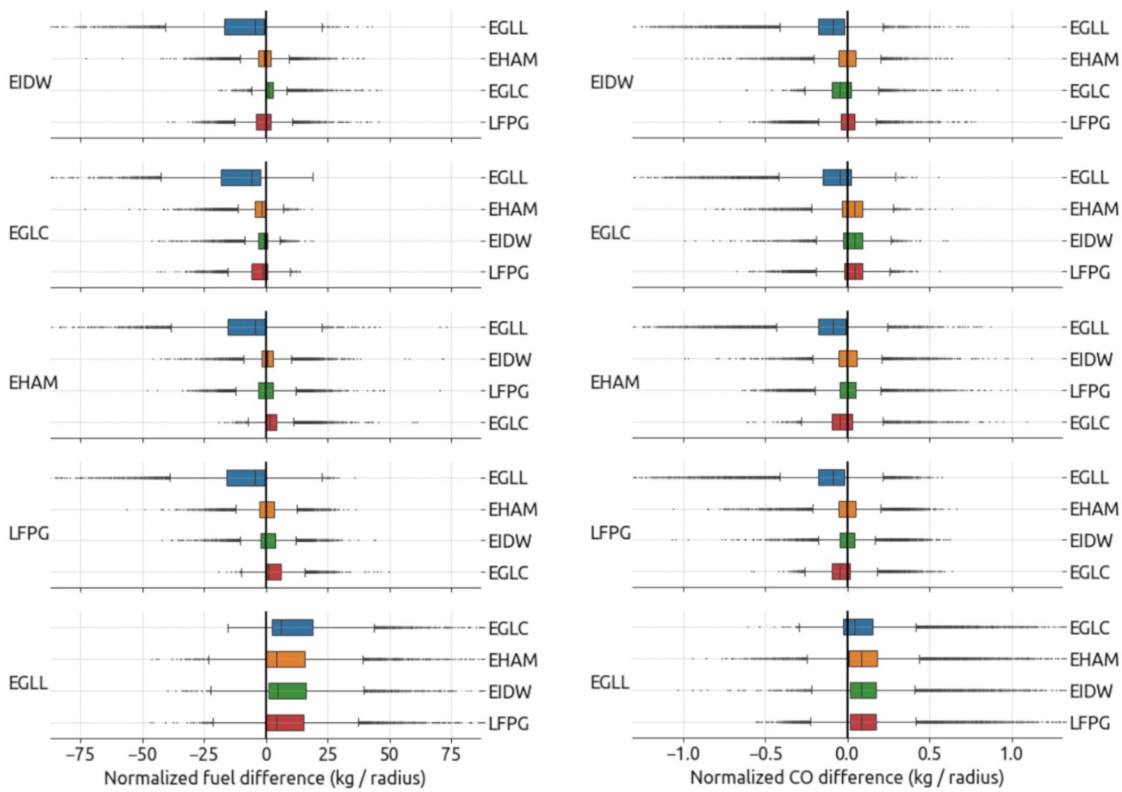

**Fig 14. Distributions of fuel consumptions and CO emissions.**

found positive effects of continuous descents and point merge patterns, but their impact was not as significant as with other procedures.

Finally, optimizing flight time does not necessarily result in lower fuel consumption, as fuel flow tends to be larger at low altitudes since fuel consumption does not directly relate to the emissions of some pollutants such as CO and HC. Future works to improve the assessment of environmental impact, a fair comparison between airports, and the design of new procedures should probably be designed with simulation tools in order to evaluate procedures that are not implemented for a certain airport.

**Table 5. Average excess emission of Airport 2 compared to Airport 1 in kilogram for 50 nm.**

| Airport 1 | Airport 2 | $CO_2$ | $H_2O$ | $SO_X$ | $NO_X$ | CO | HC |
|---|---|---|---|---|---|---|---|
| EIDW | EGLC | 149 | 58 | 0.04 | 1.01 | -1.6 | -0.11 |
| EHAM | EIDW | 193 | 75 | 0.05 | 0.55 | 0.56 | -0.02 |
| EHAM | EGLC | 332 | 129 | 0.09 | 1.53 | -1.07 | -0.14 |
| LFPG | EHAM | 1137 | 444 | 0.3 | 0.63 | 0.01 | 0.05 |
| LFPG | EIDW | 1315 | 513 | 0.35 | 1.15 | 0.42 | 0.01 |
| LFPG | EGLC | 1455 | 568 | 0.39 | 2.13 | -1.21 | -0.11 |
| EGLL | EGLC | 1692 | 661 | 0.45 | 5.58 | 5.57 | 0.45 |
| EGLL | EHAM | 1362 | 532 | 0.36 | 4.06 | 6.7 | 0.59 |
| EGLL | EIDW | 1550 | 605 | 0.41 | 4.6 | 7.13 | 0.55 |
| EGLL | LFPG | 247 | 96 | 0.07 | 3.47 | 6.77 | 0.56 |

**Table 6. Average excess emission of Procedure 2 compared to Procedure 1 in kilogram for 50 nm.**

| Airport | Procedure 1 | Procedure 2 | $CO_2$ | $H_2O$ | $SO_X$ | $NO_X$ | CO | HC |
|---------|-------------|-------------|--------|--------|--------|--------|-----|-----|
| EGLL | holding | other | 1691 | 660 | 0.45 | 5.21 | 5.75 | 0.47 |
| EGLL | CDO | other | -1155 | -451 | -0.31 | -3.59 | -3.85 | -0.3 |
| EHAM | holding | other | 998 | 390 | 0.27 | 2.63 | 4.83 | 0.5 |
| EHAM | CDO | other | -289 | -112 | -0.08 | -0.72 | -1.65 | -0.15 |
| LFPG | holding | other | 1067 | 416 | 0.28 | 1.57 | 3.36 | 0.35 |
| LFPG | CDO | other | -535 | -209 | -0.14 | -1.12 | -0.3 | -0.01 |
| EGLC | holding | other | 801 | 312 | 0.21 | 1.3 | 5.65 | 0.54 |
| EGLC | point merge | other | 363 | 142 | 0.1 | 0.58 | 2.61 | 0.24 |
| EGLC | holding | point merge | 459 | 179 | 0.12 | 0.75 | 3.24 | 0.32 |
| EGLC | CDO | other | -82 | -32 | -0.02 | -0.11 | -0.76 | -0.09 |
| EIDW | holding | other | 1079 | 421 | 0.29 | 2.67 | 5.48 | 0.56 |
| EIDW | point merge | other | 489 | 191 | 0.13 | 1.18 | 2.42 | 0.22 |
| EIDW | holding | point merge | 685 | 267 | 0.18 | 1.76 | 3.28 | 0.36 |
| EIDW | CDO | other | -519 | -202 | -0.14 | -1.26 | -3.15 | -0.31 |

## Author Contributions

**Conceptualization:** Xavier Olive, Junzi Sun, Luis Basora, Enrico Spinielli.

**Data curation:** Xavier Olive, Junzi Sun, Luis Basora.

**Formal analysis:** Xavier Olive, Junzi Sun.

**Investigation:** Luis Basora, Enrico Spinielli.

**Methodology:** Xavier Olive, Junzi Sun, Enrico Spinielli.

**Resources:** Xavier Olive.

**Software:** Xavier Olive, Junzi Sun, Luis Basora.

**Validation:** Xavier Olive, Luis Basora.

**Visualization:** Xavier Olive, Junzi Sun.

**Writing – original draft:** Xavier Olive, Junzi Sun.

**Writing – review & editing:** Xavier Olive, Junzi Sun, Luis Basora, Enrico Spinielli.

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
