## [Decision Letter · Decision Letter 0]

21 Mar 2023

PONE-D-23-01617Environmental inefficiencies for arrival flights at European airportsPLOS ONE

Dear Dr. Sun,

Thank you for submitting your manuscript to PLOS ONE. After careful consideration, we feel that it has merit but does not fully meet PLOS ONE’s publication criteria as it currently stands. Therefore, we invite you to submit a revised version of the manuscript that addresses the points raised during the review process.

We look forward to receiving your revised manuscript.

Kind regards,

Simone Lolli

Academic Editor

PLOS ONE

2. We note that Figure 3 and 4 in your submission contain [map/satellite] images which may be copyrighted. All PLOS content is published under the Creative Commons Attribution License (CC BY 4.0), which means that the manuscript, images, and Supporting Information files will be freely available online, and any third party is permitted to access, download, copy, distribute, and use these materials in any way, even commercially, with proper attribution. For these reasons, we cannot publish previously copyrighted maps or satellite images created using proprietary data, such as Google software (Google Maps, Street View, and Earth). For more information, see our copyright guidelines: http://journals.plos.org/plosone/s/licenses-and-copyright.

a. You may seek permission from the original copyright holder of Figure 3 and 4 to publish the content specifically under the CC BY 4.0 license. 

Additional Editor Comments:

Dear authors, both reviewers agree that the manuscript is clear and well written. Some minor changes are needed before publication. Please take into consideration all the suggestions/comments in drafting the new version

Reviewers' comments:

Reviewer's Responses to Questions

**Comments to the Author**

1. Is the manuscript technically sound, and do the data support the conclusions?

Reviewer #1: Yes

Reviewer #2: Yes

2. Has the statistical analysis been performed appropriately and rigorously? 

Reviewer #1: Yes

Reviewer #2: Yes

3. Have the authors made all data underlying the findings in their manuscript fully available?

Reviewer #1: Yes

Reviewer #2: Yes

4. Is the manuscript presented in an intelligible fashion and written in standard English?

Reviewer #1: Yes

Reviewer #2: Yes

5. Review Comments to the Author

Reviewer #1: Accurate article from a pilot perspective with correct aviation terminology. It indicates how different descent patterns influence emissions and fuel consumptions. However some airports like EGLL presents constraints - I.e. London TMA airspace - which make the use of CDO not practical, preventing the optimisation of emissions and fuel consumption. An useful recommendation to include, which may go beyond the scope of the article but provide some practical implications, could indicate how airports and ATC can improve the design of descent patterns taking into account the various constraints, peculiar of each airport and airspace.

Reviewer #2: This manuscript is well written and easy to understand. Each section has the appropriate amount of detail, and the figures & tables are well made. I have the following comments to help further improve the manuscript.

1. Page 7, line 3 "merge point with a DIRECT when spacing is obtained." – Please clarify what DIRECT is in this case

2. Page 8, last paragraph, line 3 "area of interest, 50nm by default, 60nm for EGLC, 90nm for LFPG". How are these sizes determined?

3. Page 9, Fig 6 – The boxplots show quite a few outliers, especially at EGLL airport. These outliers seem to have very large values for duration, normalized distance, fuel consumption, and emissions. What are the reasons for these outliers?

4. Page 9, Section 4.2, paragraph 2 "The calculation of fuel and emissions are all based on the mass that is 90 % of the maximum landing weight for the specific aircraft type." – Is this a reasonable assumption? Please add some more details or a reference to justify this assumption.

5. Page 10, Fig 7 – There seem to be 4 (1 point merge, 3 non-point merge) outliers that deviate from the general trend. These seem to have very low normalized fuel consumption even though their distance is high. Please add a discussion of these outliers in the manuscript text

6. Page 10, 2nd paragraph below Fig 7 "p-value ≪ 0.0" – Please check if there a typo here. For t-tests, the p value is always positive so it cannot be ≪ 0.0.

6. PLOS authors have the option to publish the peer review history of their article (what does this mean?). If published, this will include your full peer review and any attached files.

Reviewer #1: **Yes: **Matteo Taddei

Reviewer #2: No

---

## [Author Response · Author response to Decision Letter 0]

12 Apr 2023

We would like to thank the reviewers for their comments. Please see the attached response letter with our responses.

---

## [Decision Letter · Decision Letter 1]

8 Jun 2023

Environmental inefficiencies for arrival flights at European airports

PONE-D-23-01617R1

Dear Dr. Sun,

We’re pleased to inform you that your manuscript has been judged scientifically suitable for publication and will be formally accepted for publication once it meets all outstanding technical requirements.

Kind regards,

Simone Lolli

Academic Editor

PLOS ONE

Additional Editor Comments (optional):

Reviewers' comments:

Reviewer's Responses to Questions

**Comments to the Author**

1. If the authors have adequately addressed your comments raised in a previous round of review and you feel that this manuscript is now acceptable for publication, you may indicate that here to bypass the “Comments to the Author” section, enter your conflict of interest statement in the “Confidential to Editor” section, and submit your "Accept" recommendation.

Reviewer #2: All comments have been addressed

2. Is the manuscript technically sound, and do the data support the conclusions?

Reviewer #2: (No Response)

3. Has the statistical analysis been performed appropriately and rigorously? 

Reviewer #2: (No Response)

4. Have the authors made all data underlying the findings in their manuscript fully available?

Reviewer #2: (No Response)

5. Is the manuscript presented in an intelligible fashion and written in standard English?

Reviewer #2: (No Response)

6. Review Comments to the Author

Reviewer #2: (No Response)

7. PLOS authors have the option to publish the peer review history of their article (what does this mean?). If published, this will include your full peer review and any attached files.

Reviewer #2: No

---

## [Editor Report · Acceptance letter]

13 Jun 2023

PONE-D-23-01617R1 

Environmental inefficiencies for arrival flights at European airports 

Dear Dr. Sun:

I'm pleased to inform you that your manuscript has been deemed suitable for publication in PLOS ONE. Congratulations! Your manuscript is now with our production department. 

Kind regards, 

on behalf of

Dr. Simone Lolli 

Academic Editor

PLOS ONE